# Bcl-2 Modulation in p53 Signaling Pathway by Flavonoids: A Potential Strategy towards the Treatment of Cancer

**DOI:** 10.3390/ijms222111315

**Published:** 2021-10-20

**Authors:** Noor Rahman, Haroon Khan, Asad Zia, Asifullah Khan, Sajad Fakhri, Michael Aschner, Karim Gul, Luciano Saso

**Affiliations:** 1Department of Biochemistry, Abdul Wali Khan University Mardan, Mardan 23200, Pakistan; noorbiochemist@gmail.com (N.R.); asad.zia67337@gmail.com (A.Z.); asifullah111@gmail.com (A.K.); 2Department of Pharmacy, Abdul Wali Khan University Mardan, Mardan 23200, Pakistan; 3Pharmaceutical Sciences Research Center, Health Institute, Kermanshah University of Medical Sciences, Kermanshah 6734667149, Iran; pharmacy.sajad@yahoo.com; 4Department of Molecular Pharmacology, Albert Einstein College of Medicine, Bronx, NY 10463, USA; michael.aschner@einsteinmed.org; 5Department of Biotechnology, Abdul Wali Khan University Mardan, Mardan 23200, Pakistan; biotechdms@gmail.com; 6Department of Physiology and Pharmacology, “Vittorio Erspamer” Sapienza University of Rome, 00185 Rome, Italy

**Keywords:** flavonoids, cancer, p53, Bcl-2, pharmacology, therapeutic target, signaling pathway

## Abstract

Cancer is a major cause of death, affecting human life in both developed and developing countries. Numerous antitumor agents exist but their toxicity and low efficacy limits their utility. Furthermore, the complex pathophysiological mechanisms of cancer, serious side effects and poor prognosis restrict the administration of available cancer therapies. Thus, developing novel therapeutic agents are required towards a simultaneous targeting of major dysregulated signaling mediators in cancer etiology, while possessing lower side effects. In this line, the plant kingdom is introduced as a rich source of active phytochemicals. The secondary metabolites produced by plants could potentially regulate several dysregulated pathways in cancer. Among the secondary metabolites, flavonoids are hopeful phytochemicals with established biological activities and minimal side effects. Flavonoids inhibit B-cell lymphoma 2 (Bcl-2) via the p53 signaling pathway, which is a significant apoptotic target in many cancer types, hence suppressing a major dysregulated pathway in cancer. To date, there have been no studies reported which extensively highlight the role of flavonoids and especially the different classes of flavonoids in the modulation of Bcl-2 in the P53 signaling pathway. Herein, we discuss the modulation of Bcl-2 in the p53 signaling pathway by different classes of flavonoids and highlight different mechanisms through which this modulation can occur. This study will provide a rationale for the use of flavonoids against different cancers paving a new mechanistic-based approach to cancer therapy.

## 1. Introduction

Cancer is the primary cause of death all over the world and millions of individuals are diagnosed with cancer annually, foremost at later life stages [1]. Worldwide, an estimated 19.3 million new cancer cases (18.1 million excluding nonmelanoma skin cancer) and almost 10.0 million cancer deaths (9.9 million excluding nonmelanoma skin cancer) occurred in 2020. The global cancer burden is expected to be 28.4 million cases in 2040, a 47% rise from 2020, with a larger increase in transitioning (64% to 95%) versus transitioned (32% to 56%) countries due to demographic changes, although this may be further exacerbated by increasing risk factors associated with globalization and a growing economy [2].

Numerous antitumor agents are being used to combat cancer; however, they cause toxicity which limits their administration [3,4]. Due to serious side effects and poor prognosis of available cancer therapies, such as radiotherapy, surgery, and chemotherapy, their therapeutic effect is limited [5]. Medical plants are notable sources for novel anticancer drug discovery. Approximately 50% of anticancer drugs approved between the 1940s and 2006 are derived from natural compounds [6]. Among them, flavonoids are natural compounds with promising biological activities and health benefits. These plant-derived secondary metabolites have been shown to target multiple dysregulated pathways in cancer [7,8].

Apoptosis plays a critical role in preventing the progression of cancer. Growing evidence has established a major role for B-cell lymphoma 2 (Bcl-2) through the p53 pathway in apoptosis and cancer phases [9,10,11]. The Bcl-2 family of proteins is comprised of Bcl-2, Bcl-xl, and Mcl-1 [12]. The initiators Bcl-2 Homology 3 (BH3) only proteins induce apoptosis by either interacting with Bax, Bak, and Bok or by binding to the anti-apoptotic proteins to liberate Bax and bak [13]. Therefore, anti- and pro-apoptotic proteins of the Bcl-2 family will drive cell survival or death. Overexpression of the Bcl-2 protein has been reported in prostate cancer, breast cancer, B-cell lymphomas, and colorectal cancer [14]. Overexpression of the Bcl-2 protein promotes cell survival and proliferation. In recent years, several BCL-2 protein inhibitors have been developed (Oblimersen, ABT-737, ABT-263, obatoclax mesylate, AT-101, and S55746), to specifically target BCL-2 protein for cancer treatments [15,16,17]. Therefore, Bcl-2 is a validated drug target, and inhibitors targeting Bcl-2 have a wide range of clinical applications.

Dietary flavonoids in p53-mediated immune dysfunctions have been linked to cancer prevention [18,19]. The current study is the first review regarding targeting Bcl-2 through the p53 pathway by flavonoids in cancer.

## 2. The Role of Ethnopharmacology in Cancer Therapy

The majority of natural products used for nutritional or therapeutic uses are derived from higher plants [20], including plant-derived phytochemicals [21,22]. Amongst these therapeutic agents, >200,000 metabolites have been isolated from the plant kingdom [23]. Plant-derived phytochemicals have been used for the treatment of various diseases for thousands of years [24]. The U.S. National Cancer Institute (NCI) first reported on the efficacy of natural products as anticancer agents in the 1950s [3]. Presently, >60% of cancer patients use the medicinal plant as suitable alternative therapies [25]. In addition, despite the increased recognition of natural products in cancer therapy presented in Table 1, only about one-fifth to one-sixth of plant species have been evaluated for medical purposes [26].

## 3. Flavonoids: Structure, Biological Activities and Health Benefits

Generally, flavonoids are divided into various sub-classes including flavonols, flavones isoflavones, flavanones, anthocyanins, chalcones, and flavan-3-ols as presented in Figure 1. The chemical structures of flavonoids possess a fifteen-carbon based skeleton, including two benzene rings (A and B) connected via a heterocyclic pyran ring (C) as presented in the structure of Quercetin in Figure 2. Differences in the substitution of the C ring generate different classes of flavonoids, while those differences in A and B rings result in individual compounds within a flavonoid class. [85,86,87].

In terms of their anticancer effects, flavonoids target several signaling mediators to exert their therapeutic effects. Down-regulation of mutant p53, arresting the cell cycle, inhibition of tyrosine kinase/heat shock proteins, Ras protein inhibition, and binding to estrogen receptor are the common mechanisms of flavonoids in combating cancer [88]. Among the aforementioned mediators, Bcl-2 and p53 seem to play crucial roles in suppressing cancer.

## 4. Bcl-2 in Cancer Etiology

The Bcl-2 protein is a member of the Bcl-2 family and an oncogene resulting from chromosome translocation, which causes malignant lymphomagenesis [89]. In the early 1990 s, Bcl-2 was identified as an anti-apoptotic protein that prevents cell death. On the other hand, Bcl-2 Associated X (Bax) protein, which has similar sequence and structure homology to Bcl-2 and heterodimerizes with it, induces apoptosis. Accordingly, Bcl-2 family proteins are classified into pro and anti-apoptotic proteins [90]. These proteins affect the mitochondrial outer membrane permeabilization, allowing for the release of cytochrome C and DIABLO (second mitochondria-derived activator of caspases or SMAC) from the intermembrane mitochondrial space into the cytosol. The released cytochrome C binds to apoptosis protease–activating factor 1 (Apaf-1) and caspases [91]. Cytochrome C and Apaf-1 also stimulate caspase-3, c-Jun N-terminal Kinase (JNK), and p53 to inhibit cyclin-D in turn activating apoptotic pathways [92].

It has been earlier reported that the Bcl-2 expression of malignant/normal myeloid lineage cells produced significant effects [93]. Overexpression of the Bcl-2 gene is inherent to various cancers, for example, 90% of colorectal adenocarcinomas, 80% of undifferentiated nasopharyngeal cancers, 70% of breast adenocarcinomas and chronic lymphocytic leukaemias, 60% of gastric cancers, 30–60% of prostate cancers, and in varying percentages of melanomas, blastomas, and kidney cancers [94].

## 5. p53 in Cancer: Its Association with Apoptosis and Bcl-2

Approximately half of the cancers are associated with inactivated p53 [95]. Major biological functions of p53 include apoptosis, senescence, angiogenesis, cell cycle regulation, cellular differentiation, and DNA metabolism [96]. p53 acts as a sensor and restricts cell propagation under destructive conditions, including oncogene signals, hypoxia, ribosome dysfunction, DNA damage, and nutrient deprivation [97]. Accordingly, during low-level stress, p53 affords pro-survival and protective responses, including antioxidant responses, cell-cycle arrest, and DNA repair, to maintain genome integrity/viability [97]. In contrast, following cell exposure to potent stress signals, p53 provides irreversible programs of senescence or apoptosis [98,99]. p53 induces apoptosis by activating the Bax gene (a key member of the Bcl-2 family) [100]. In turn, Bax binds to Bcl-2, thereby activating the production of apoptotic mediators (e.g., caspase 3/9 and cytochrome C). Thus, targeting Bcl-2 through p53 offers efficient means for combating cancer.

## 6. Bcl-2 Inhibition Passes through the p53 Pathway by Flavonoids

A linkage exists between the activity of Bcl-2 and p53. Considering the potential role of flavonoids (e.g., flavonols, flavones, isoflavones, flavanones, chalcones, anthocyanins, and catechins) in the modulation of Bcl-2 through the p53 pathway, they could be promising agents in the treatment of cancer (Figure 1).

### 6.1. Flavonols

Quercetin is a bioflavonoid found in abundance in grapes, citrus fruits, berries, and onions. In human breast cancer (MCF-7) cells, quercetin (Figure 2) treatment effectively suppressed the cell proliferation in both the dose and time-dependent manner. Quercetin also significantly reduced Bcl-2 expression levels while increased Bax, resulting in the induction of apoptosis [27]. Quercetin treatment of prostate cancer-induced rats significantly increased the levels of antioxidant enzymes. The expression levels of Akt and anti-apoptotic protein Bcl-2 were downregulated and caspase-3 protein expression was upregulated. Moreover, quercetin down-regulated cell proliferation, viability and thus acted as a chemopreventive agent against prostate cancer in the rat model [28]. Granado-Serrano et al., 2006 reported quercetin-induced apoptosis in HepG2 cell line and evaluated the modulation and expression of Bcl-x and Bax. Bcl-x_L_ has been identified as a caspase substrate and the product of Bcl-x_L_ cleavage, Bcl-x_S_, has a pro-apoptotic function. The level of Bcl-x_S_ was elevated at all quercetin concentrations compared with controls after 18 h of incubation. Quercetin reduced the Bcl-x_L_:Bcl-x_S_ ratio, which reached a minimum at 50 µmol/L. Bax has been demonstrated to translocate from the cytoplasm to the outer mitochondrial membrane, where it forms holes and mediates apoptosis. Western blot analysis of mitochondrial and cytoplasmatic fractions revealed that after 18 h of quercetin treatment, translocation of Bax to the mitochondria increased to its greatest level at 50 mmol/L, decreased at 75 mmol/L, and returned to control levels at 100 mmol/L [101].

The kaempferol (Figure 2) caused a marked anti-cancer effect in MCF-7 breast cancer cell lines mediated by down-regulation of Bcl-2 expression, accompanied by the overexpression of Bax protein and thus produced apoptosis [29]. Kaempferol therapy slowed the progression of tumor xenografts by downregulating the proteins cyclin B1 and Cdk1. Furthermore, kaempferol treatment reduced the level of Bcl-2 while upregulated Bax expression consequently acting as chemopreventive agent [30]. Miquelianin (Figure 2) (Quercetin-3-O-glucuronide) has a protective role against 1-methyl-4-phenylpyridinium-induced neurotoxicity. MTT assay indicated miquelianin significantly inhibited apoptosis, which was accompanied by a decrease in PARP cleavage. Additionally, it attenuated MPP-induced intracellular ROS with the decrease in Bax/ Bcl-2 ratio [83].

Galangin (Figure 2) was isolated from the rhizome of *Alpinia officinarum* has anticancer effects against many cancer cells such as liver, lung, breast, and esophageal cancer. It inhibited cell proliferation, induced apoptosis evident from the reduced Bcl-2 and higher cleaved caspase-3 expressions [4].

Casticin (Figure 2) induced apoptotic cell death in human lung cancer cells and caused the activation of multiple apoptotic proteins such as procaspase-9 and procaspase-3. Additionally, casticin down-regulated Bcl-XL and upregulated Bax, and also increased death receptor 5 (DR5) expression levels [31]. In another study, casticin caused cycle arrest at G0/G1 phase and induced apoptosis by increasing the expression of Bax and p27 proteins and down-regulating Bcl-2 expression in human gallbladder cancer cells [102].

Fisetin (Figure 2) treatment of plasma cancer cells (U266) [103] and human non-small cell lung cancer cell lines (NCI-H460) [32] promoted the activation of caspase-3, upregulated Bax, Bim and Bad proteins expression while down-regulated Mcl-1L and Bcl-2 expression. Consistently, morin, a bioflavonoid found in the mulberry that acts as an apoptotic inducing agent in multiple human cancers. Morin (Figure 2) induced the upregulation of the Fas receptor and activated caspase-8, caspase-9, and caspase-3 in human colon cancer (HCT-116) cells. It additionally caused a mitochondrial potential loss, activated Bax protein, inhibited Bcl-2 and enhanced the generation of reactive oxygen species (ROS) [33]. Morin exhibited a protective effect in myocardial ischemia-reperfusion injury (MIRI) by increasing cell viability and enhanced the regaining of heart function in rats. Moreover, morin treatment prevented the decrease of mitochondrial membrane potential and reduced the levels of cytochrome c, caspase-9, and caspase-3. Furthermore, morin treatment significantly down-regulated the expression of Bax while upregulated the expression of Bcl-2 [34].

Tamarixetin (Figure 2) showed cytotoxicity towards leukemic cells, inhibiting cancer cells proliferation by enhancing apoptotic activity and blocking cell cycle progression accompanied by the increase in p21 and cyclin B1. In addition, tamarixetin induced an increase in Bax expression and decreased Bid expression. Apoptosis was caused due to the release of cytochrome C, activation of caspases and cleavage of poly ADP-ribose polymerase (PARP) [35]. From another point of view, oxidative stress causes mitochondrial membrane loss and may cause myocardial damage. H_2_O_2_ is a Reactive Oxygen Species (ROS) that activates caspase-3 and increases Bax expression while decreasing Bcl-2 expression and causing cell death in rat H9c2 cell lines. In this line, rutin treatment reversed this process by inhibiting Bax and caspase-3 while increases Bcl-2 expression and act as an anti-apoptotic agent preventing myocardial damage caused by oxidative stress [36].

Icariin (Figure 2) inhibited the growth of many tumor cells [104]. In human hepatoma (SMMC-7721) cell lines, icariin initiated the mitochondrial-dependent apoptotic pathway by increasing the Bax/Bcl-2 ratio, dysfunctioning of mitochondrial membrane potential to make the release of cytochrome C and activating caspase cascade. Icariin also triggered ROS generation in SMMC-7721 cells [37].

### 6.2. Flavones

Acacetin (Figure 3) activated apoptosis in human breast cells (MCF-7) by activating caspase-7 and mitochondrial-mediated death signaling. Acacetin treatment significantly inhibited Bcl-2 expression while increased the expression levels of Bax. Acacetin increased the release of apoptosis-inducing factor (AIF) into the cytoplasm leading to ROS generation and subsequently induces apoptosis in hepatoma cell lines (SMMC-7721) [38]. Amongst other flavones, wogonin (Figure 3) treatment increased the apoptotic activity of SMMC-7721 by modulating the expression of Bcl-2 protein and Bax protein in a time and dose-dependent manner. The expression of Bcl-2 protein was significantly reduced while the expression of Bax protein was increased [39].

In human lung cancer (A549) cell lines, apigenin (Figure 3) treatment caused cytotoxicity by inducing DNA damage and decreasing cell viability in a dose-dependent manner. Apigenin induced apoptosis accompanied by the increase in the expression of Bax protein and inhibited Bcl-2 protein level leading to the disruption of mitochondrial membrane, which caused the release of cytochrome C and Endo G and induced the activation of caspases [40]. In hepatocellular carcinoma cells (HCC), chrysin (Figure 3) treatment increased the expression of pro-apoptotic agents such as p53, Bad, Bax and Bak proteins and reduced the expression levels of anti-apoptotic agents such as Bcl-2 protein which resulted in the induction of apoptosis. Chrysin also suppressed the viability of hepatocellular carcinoma cells dose-dependently [41].

Luteolin (Figure 3) exhibited anticancer and antitumor effects against many types of cancers such as brain, lung, breast, prostate and pancreatic cancers [105]. The pro-apoptotic and anti-proliferative activities of luteolin were analyzed in vitro in colon adenocarcinoma (HCT-15) cells. Luteolin treatment inhibited Wnt/β-catenin signaling pathway and induced cell cycle arrest at the G2/M phase. It suppressed Bcl-2 expression levels and increased the expression of Bax and caspase-3 [42]. In human breast cancer lines MCF-7, a combination of baicalin and baicalein (Figure 3) showed remarkable anti-proliferative activity in a time and dose-dependently. This combinatorial treatment activated the caspase cascade, upregulated p53 and Bax expression while down-regulated Bcl-2 expression which is associated with the activation of ERK/p38 mitogen-activated protein kinase (MAPK) pathway [43]. In HeLa cell lines, eupatorin (Figure 3) treatment caused cell cycle arrest at the G2/M phase and then induced cell death by inhibiting cyclin D1, initiated the cleavage of caspase cascade, enhanced p53, p21 and Bax expression levels through activating both p53 dependent and independent pathways [44].

Sinensetin (Figure 3) increased caspases and PARP expression levels and induced autophagy in human T-cell lymphoma cells by activating ROS/terminal kinase and inhibiting Akt/mTOR signaling pathways [45]. Nobiletin (Figure 3) induced cell death in human breast cancer MCF-7 cells by modulating the expression of Bax and Bcl-2 proteins. Nobiletin upregulates the apoptotic inducing proteins Bax and p53 while down-regulate the antiapoptotic Bcl-2 protein in the breast cancer cell line (MCF-7). It also blocked cell migration by inhibiting MMP-2 and MMP-9 proteins [46].

The eupatilin (Figure 3) treatment has reversed the apoptosis in PC12 cells by enhancing Bcl-2 expression, inhibiting Bax protein and inactivating caspase-3 to prevent oxidative stress-induced neuronal injury in PC12 cell lines [47]. Similarly, eupatilin pretreatment attenuated myocardial ischemia/reperfusion injury by inhibiting apoptosis and reducing oxidative stress through the activation of the Akt/glycogen synthase kinase-3β (GSK-3β) pathway [48].

Vitexin (Figure 3) protected against heart failure in rats by inhibiting oxidative stress-induced myocardial apoptosis through decreasing Bax and increasing Bcl-2 protein expression [50]. Similarly, it has been found that vitexin induced apoptotic activity and decreased Bcl-2/Bax expression ratio while increased the expression of cleaved caspase-3 in human non-small cell lung cancer A549 cells. Additionally, it induced the release of cytochrome C into the cytosol from the mitochondria leading towards the loss of mitochondrial membrane potential [49].

Pectolinarigenin (PG) (Figure 3) blocked osteosarcoma cells proliferation, induced cell death and decreased the level of cyclin D1, Survivin, Bcl-2 and Bcl-xL proteins [51]. Wu et al. investigated the protective activity of PG in a rat model of spinal cord injury, PG effectively enhanced functional recovery and inhibited apoptosis in neuronal cells by down-regulating the activated caspase proteins and PARP, decreasing Bax expression, and increasing Bcl2 expression [52].

In human prostate cancer, morusin (Figure 3) inactivated STAT3 signaling and caused apoptosis by inhibiting Bcl-2, Bcl-xL and surviving. Furthermore, it has been found that morusin exerted growth inhibitory effects on HCC cells both in vitro and in vivo. Additionally, it induced apoptosis accompanied by the increased active caspase-3 and decreased Bcl-2 expression [53].

Vicenin-2 (Figure 3) reversed the diethyl nitrosamine-induced liver carcinoma in rats by potentially inhibiting the production of ROS, decreasing the liver weight, and reducing cellular changes in the liver which were previously induced by the diethylnitrosamine in rats. Furthermore, vicenin-2 downregulated the expression of anti-apoptotic proteins Bcl-xL and Bcl-2 while upregulated the expression of pro-apoptotic protein Bax and cleaved caspase-3 [54]. It has been reported that vicenin-2 stimulated significant cell cycle arrest at the G2/M phase and also induced apoptotic cell death in HT-29 cells [55]. Moreover, vicenin-2 treatment upregulated the expression of caspase-3 and Bax, increased the dysfunction of the mitochondrial membrane potential whereas down-regulated the Bcl-2 expression.

Hydroxygenkwanin (Figure 3) in combination with apigenin inhibited brain tumor cells proliferation through upregulating TNF-α levels, activating caspase-3, caspase-8, and down-regulating Bcl-2 [55]. It has been investigated that hydroxygenkwanin along with kaempferol showed both cytotoxic and antioxidative potential against HepG2 cell lines [106]. This may be due to the fact that hydroxygenkwanin has multiple hydroxyls that may donate hydrogen and act as an antioxidant in forming phenoxyl radicals, therefore induce cytotoxicity.

### 6.3. Isoflavones

In this class of phytochemicals, puerarin (Figure 4A) act as a protective agent against ROS-induced apoptosis by decreasing Bax/Bcl-2 ratio and apoptosis, as well as preventing neuronal disorders such as Alzheimer’s disease [107] and PC12 cancer cells [56]. Puerarin treatment of HCC cell lines increased the phosphorylation and activation of MAPK and act as an anticancer agent by exhibiting pro-apoptotic activities [108]. Combinatorial treatment of genistein and hypericin (Figure 4A) in human breast cancer cells resulted in the reduction of Bcl-2 expression while an increase in Bax expression suppressed Akt and ERK1/2 phosphorylation [57].

Daidzein (Figure 4A) initiated cell death in SK-HEP-1 cells by enhancing Bak expression and inhibiting Bcl-2 and Bcl-xL proteins through mitochondrial-mediated apoptosis [58]. While ultraviolet radiation damages the skin extracellular matrix and induces cell death, tectorigenin (Figure 4A) treatment decreased the levels of ROS, increased anti-apoptotic Bcl-2 protein expression, overexpressed glutathione and catalase and inhibited skin cells death [59]. Jaceosidin (Figure 4A) induced cell cycle arrest at the G2/M phase, upregulated the expression levels of p53 and Bax proteins and caused the loss of mitochondrial membrane potential by releasing cytochrome C and activating caspase-3 in U87 cell lines [109]. Isoangustone A (Figure 4A) induced apoptosis and activated the caspase cascade cleavage while down-regulated Bcl-2 protein in human colorectal adenocarcinoma (SW480) cells [60]. The Biochanin A was initially found in the *Trifolium pretense* L.(clover) plant and was extracted from the stems and leaves. Biochanin A exhibits significant anticancer and antioxidant activities. It induced apoptosis, block metastasis, and caused cell cycle arrest by targeting multiple signaling pathways of cancer [110].

### 6.4. Flavanones

Several flavanones have also shown potential anticancer effects through Bcl-2 modulation in the p53 signaling pathway. Among them, hesperetin (Figure 4B) treatment causes cell cycle arrest at G1-phase in MCF-7 cell lines, through downregulating the cyclins and upregulating p21. Additionally, hesperetin enhances the binding of CDK4 with p21, which suggests the fact that hesperitin is involved in anticancer pathways [111]. In addition, hesperetin induced apoptosis in gastric cancer cells by activating the mitochondrial signaling pathway which caused the upregulation of Bax protein expression and down-regulating Bcl-2 expression [112]. Hesperidin (Figure 4B) also caused cell death in human colon cancer cells through caspase-3 activation. It significantly enhanced Bax expression and reduced the expression of Bcl-2 [61]. Liquiritigenin (Figure 4B) effectively enhanced cell viability and inhibited palmitate-induced apoptosis by decreasing the cleavage of caspases and PARP while upregulating Bcl-2 expression [62].

Naringin (Figure 4B) treatment prevented gentamicin-induced nephrotoxicity, through a potential reduction of caspase-3, p53 and Bax as well as enhancement of the Bcl-2 protein expression [63]. Similarly, Saralamma et al. showed anticancer effects of poncirin (Figure 4B). They reported that poncirin treatment increased the expression of death receptors Fas Ligand (FasL) protein in human gastric cancer cells. Additionally, it induced the activation of caspase-8 and caspase-3 and cleavage of PARP [64]. Kuarinone (Figure 4B), norkurarinol and 2′ methoxy kurarinone are kushen flavonoids that inhibit the growth of many cancers cell lines such as A549, SPC-A-1 and NCI-H46, respectively [113]. Kurarinone inhibited the proliferation of A549 cell lines and decreased Bcl-2/Bax levels while activating the caspase-9 and caspase-3 [65].

### 6.5. Chalcones

The isoliquiritigenin (Figure 5) have shown a significant anticancer effect through decreasing the production of prostaglandin E2 (PGE2) and nitric oxide (NO) in mice macrophages. This decrease in PGE2 was influenced by the downregulation of cyclooxygenase-2 expression, Bcl-2 and decreasing in NO which was influenced by the low expression of inducible nitric oxide synthase (iNOS) [114,115]. Isoliquiritigenin treatment suppressed the growth of abnormal cells and induced apoptotic activity in mouse and human colon carcinoma cells [66]. In Ca Ski cells, isoliquiritigenin down-regulated the expression of HPV16 E6 which is associated with the increased levels of p53 and p21, enhanced Bax expressions and decreased Bcl-2 expressions and sequentially activated caspase cascade by cleaving caspase-9, caspase-3 and PARP [116].

Liquiritin, isoliquirigenin and isoliquiritin (Figure 5A) when applied in combination, against lung cancer cells, increased cytotoxic capacity and upregulated the p53 and p21 proteins, also downregulated the expression of MDM2, Bcl-2, p-Akt proteins through p53 dependent signaling pathway. Additionally, it inhibited cell cycle arrest at the G2/M phase [67]. Licochalcone A (Figure 5A) induced apoptosis by modulating Bcl-2 protein expression and decreased Bcl-2/Bax ratio in MCF-7 and HL-60 cell lines [68]. Licochalcone B (Figure 5A) treatment reduced Bcl-2 and survivin levels, increased Bax expression, also activated caspase-3 and cleaved PARP protein [69]. Licochalcone E (Figure 5A) treatment increased the Fas ligand expression levels and increased caspase-8 proteins in FaDu cells. Moreover, Lico-E treatment increased apoptotic activity by upregulating Bax, caspase-9 and onco-suppressor p53 whereas decreases Bcl-2, consequently, induced apoptosis by both intrinsic and extrinsic signaling pathways [70].

### 6.6. Anthocyanin

A recent study has introduced anthocyanins as promising anticancer agents [8]. In this regard, malvidin (Figure 5B) upregulated p21 expression levels in human colorectal cancer [71] and its combined use with blueberry induced cell death through Bax-mediated intrinsic pathway in SCC131 cells by inhibiting Bcl-2 with an increase in Bax expression and initiating cleavage of caspases [72]. Cyanidin-3-O-β-glucopyranoside (Figure 5B) treatment of leukemia cell lines caused cell death and significantly upregulated p53 and Bax expression also down-regulated Bcl-2 expression in a time-dependent manner [73]. Pelargonidin (Figure 5B) treatment reduced the expression of Bcl-xL and Bcl-2 and increased the expression of Bid and Bax. Additionally, it enhanced p53 and p21 expression levels in human colorectal cell lines [74].

In human breast cancer cells, delphinidin (Figure 5B) treatment inhibited cell proliferation by blocking the Akt signaling pathway and inducing apoptosis by increasing Bcl-2 expression along with increasing Bax expression in a dose-dependent manner [75]. Bilberry extract (Figure 5B) (which contain delphinidin-3-O-glucoside, cyanidin-3-O-glucoside, delphinidin-3-O-rutinoside, cyanidin-3-O-galactoside, and cyanidin-3-O-rutinoside flavonoids) was applied to chronic lymphocytic leukemia which activated caspase-3, de-phosphorylated Akt and inhibited Bcl-2 and resulted in the induction of apoptosis [108]. Procyanidins B and procyanidin C (Figure 5B) are formed from the oligomers of catechin and epicatechin. When they are depolymerized in an oxidative environment, cyanidins are formed. Procyanidins extracted from *Pinus koraiensis* bark promoted apoptosis in the HeLa cell line by raising Bax protein expression and inhibiting Bcl-2 and survivin protein expression [117].

### 6.7. Flavan-3-ols

Epigallocatechin-3-gallate (EGCG) (Figure 6) is found in green tea to inhibit growth and induce apoptosis in various types of human cancer cells. EGCG inhibited gastric and hepatocarcinoma cell growth and reduced Bcl-2 expression in a time-dependent manner [77]. EGCG when applied to colon cancer cells, inhibited cyclooxygenase-2 (COX-2) and activated AMP-activated protein kinase (AMPK) accompanied by a decrease in vascular endothelial growth factor (VEGF) and Glut-1 levels [60,78]. Chemical stress damages the lens epithelium of rats and causes apoptosis through increasing Bax/Bcl-2 ratio, which could be regulated by catechin [118]. Catechin (Figure 6) and gemigliptin have anti-apoptotic effects on tacrolimus-induced renal injury in mice [119]. Their study confirmed that combination use of catechin and gemigleptin exerts anti-apoptotic effects by increasing the expression of anti-apoptotic protein Bcl-2 in tacrolimus-induced nephropathic mice. Iranian green tea extract (IGTE) contains active flavonoid catechin. Safari et al., reported that treatment of A549, PC3, and MCF-7 cell lines with IGTE induced apoptosis by increasing the levels of Bax and decreasing the expression of Bcl-2 [80].

Epicatechin (Figure 6) is one of the most abundant flavonoids found in apples, grapes, blackberries, etc. A research study suggested that Epicatechin (EC) extracted from *Euonymus alatus* had a protective effect against acute liver injury of mice by inhibiting apoptosis in hepatocytes. Western blot analysis revealed reduced expression of cleaved Caspase-3 and Bax. Their results revealed it increased the expression of Bcl-2, confirming its protective effect in liver injury [82].

## 7. Challenges and Possible Solutions to Flavonoids Therapy

The low water solubility of most flavonoids coupled with their shorter intestinal residence time as well as their lower absorption refrain humans to suffer acute toxic effects from flavonoids consumption. Although most flavonoids/phenolics are considered safe, flavonoid/phenolic therapy or chemopreventive use should be evaluated because there have been reports of toxic flavonoid-drug interactions, liver failure, hemolytic anaemia, contact dermatitis, and estrogenic-related concerns such as male reproductive health and breast cancer linked to dietary flavonoid/phenolic consumption or exposure [120]. However, the low water solubility and bioavailability of flavonoids present a potential problem for its medicinal applications [121]. Nanotechnology can serve as an efficient tool in eradicating the limitations stated above. By reducing the size of the flavonoids based nano-medicine and modifying their surface properties, the aqueous solubility and permeability through the biological membrane can be potentially increased [122]. Several novel nanotechnology-based drug delivery systems have been reported (such as niosomes, liposomes, phytosomes, and nanospheres) to potentially improve the bioavailability and therapeutic efficacy of flavonoids. The incorporation of phytomedicines (e.g., flavonoids) in these delivery systems also aid in increasing the solubility, enhancing stability and therapeutic efficacy, minimizing toxicity, improving tissue macrophage distribution, sustained delivery and protection from chemical and physical degradation [123].

## 8. Conclusions and Future Direction

Flavonoids are phytochemicals with potential biological activities and health benefits. They regulate several signaling pathways to target apoptosis, inflammation, and oxidative stress, thereby exerting potential anticancer effects. Among those signaling pathways, modulation of Bcl-2 through the p53 pathway seems to be of great importance. Flavonoids have been found to effectively modulate Bcl-2 through the p53 pathway in cancer (Figure 7).

Flavonoids are degraded by intestinal microorganisms/enzymes after being administered orally. Thus, developing innovative flavonoid delivery strategies may enhance their anticancer properties [8]. Future studies should characterize signaling pathways responsive to flavonoids and provide insight into their antitumor potential, addressing their efficacy in the treatment and management of cancers.

## Figures and Tables

**Figure 1 ijms-22-11315-f001:**
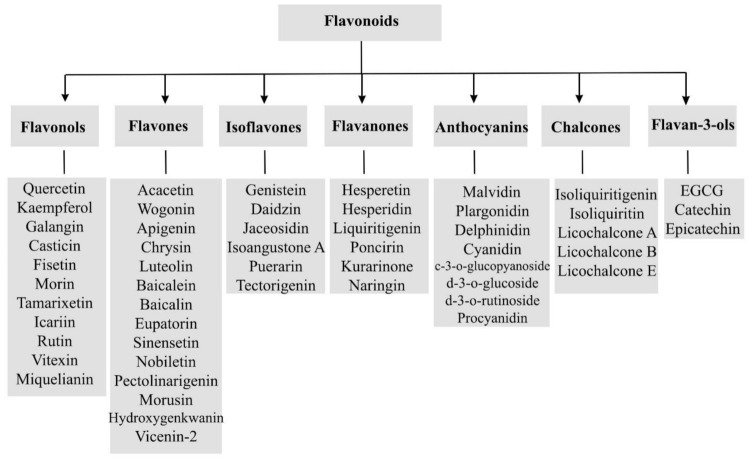
A general scheme of the classification of flavonoids.

**Figure 2 ijms-22-11315-f002:**
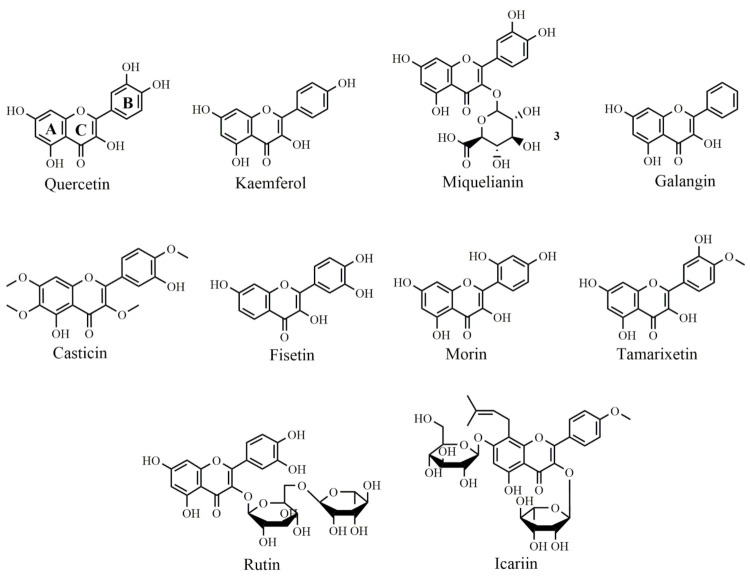
Selected chemical structures of flavonols targeting Bcl-2 in the p53 pathway. Two benzene rings (**A** and **B**) connected via a heterocyclic pyran ring (**C**).

**Figure 3 ijms-22-11315-f003:**
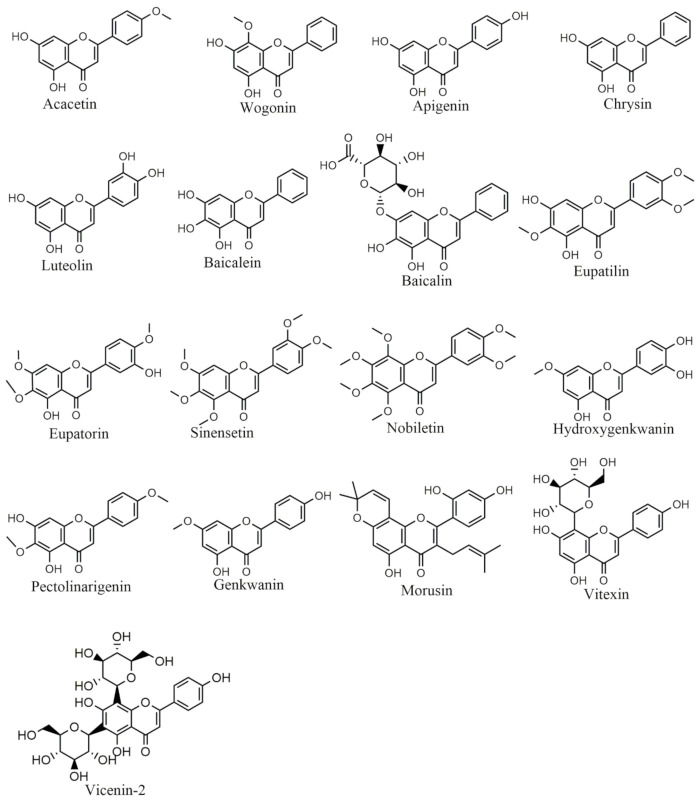
Selected chemical structures of flavones targeting Bcl-2 in the p53 pathway.

**Figure 4 ijms-22-11315-f004:**
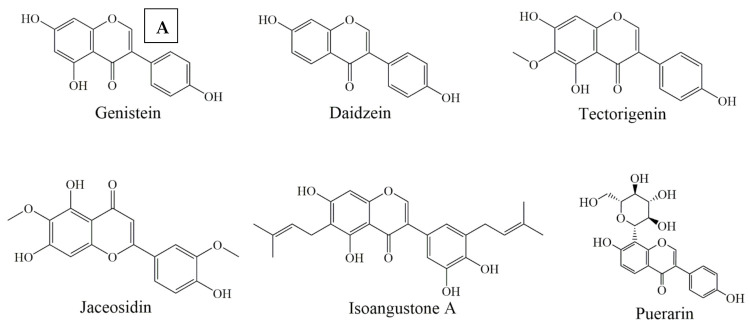
Selected chemical structures of isoflavones (**A**), and flavanones (**B**) targeting Bcl-2 in the p53 pathway.

**Figure 5 ijms-22-11315-f005:**
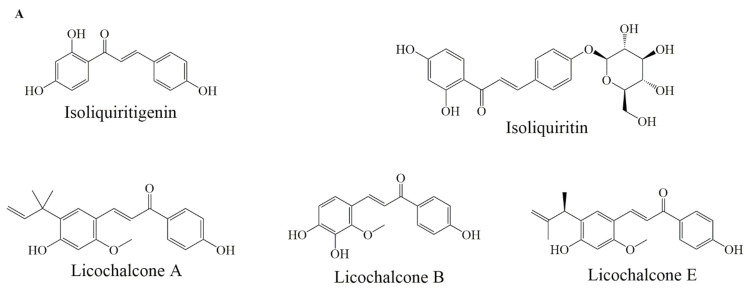
Selected chemical structures of chalcones (**A**), anthocyaninsx and catechins (**B**) that target Bcl-2 in the p53 pathway.

**Figure 6 ijms-22-11315-f006:**
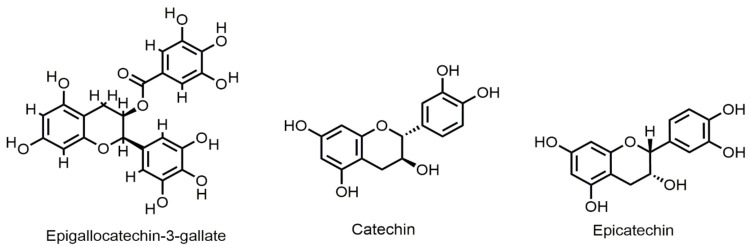
Selected chemical structures of flavan-3-ols targeting Bcl-2 in the p53 pathway.

**Figure 7 ijms-22-11315-f007:**
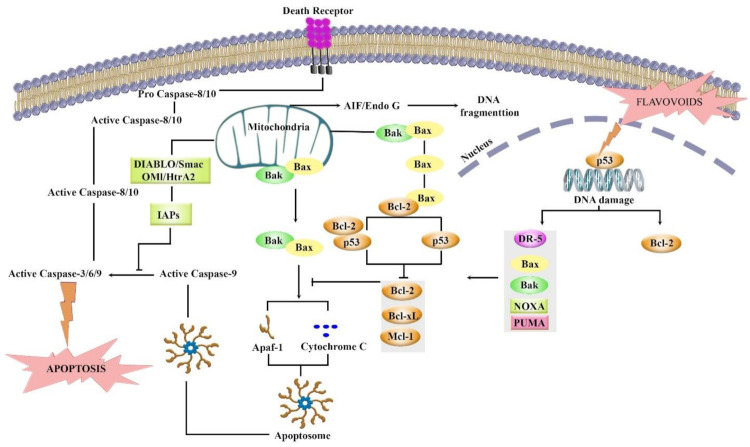
Targeting Bcl-2 in the p53 pathway by different flavonoids. It shows details of various events including up and downregulation, or expression that leads to apoptosis.

**Table 1 ijms-22-11315-t001:** The in vitro and in vivo activities of flavonoids in different cancer cell lines and their mechanism of action.

Flavonoids		Plant Source		Cell Line (s)/Cancer Model (s)	Time of Treatment	Effective Dose (In Vivo*)* or GI50, IC50, EC50 (In Vitro)	Mechanism of Action/Metabolic Effects	Reference(s)
Quercetin		*Vitis vinifera, Citrus japonica, Citrus sinensis*		Human breast cancer cell line (MCF-7)	24 h, 48 h, 72 h	50–200 μM	↓Bcl-2 ↑Bax expression	[27]
				Prostate cancer-induced rat model	3 times a week (for 16 weeks)	200 mg/kg body wt.	↓Cell proliferation ↓Akt ↓Bcl-2 ↑Caspase-3	[28]
Kaempferol		*Justicia spicigera, Acacia nilotica, Pteridium aquilinum*		Human breast cancer cell line (MCF-7)	24 h	40 μM, 80 μM	↓Bcl-2 ↑Bax expression	[29]
				SGC7901 cell-derived xenograft tumor	24 h	20 mg/kg	↓Tumor progression ↓Cdk1 ↓Cyclin B1 ↓Bcl-2 ↑Bax	[30]
Casticin		*Vitex agnus-castus*		Human non-small-cell lung carcinoma cell lines (NCI-H460, A549 and H157). Gallbladder cancer cell lines (NOZ and SGC996)	24 h	4.0 μmol/L	Activation of procaspase-9 and procaspase-3, ↓Bcl-XL ↑Bax expression ↑Death receptor 5 (DR5) expression. Cell arrest at G0/GI phase by ↑Bax ↑p27 ↓bcl-2	[31]
Fisetin		*Fragaria ananassa, Vitis vinifera, Allium cepa*		Human plasma cell line (U266), Human non-small-cell lung cancer cell line (NCI-H460)	24 h	125 μg/mL, 75 μg/mL	↑caspase-3 ↑Bax, Bim and Bad expression ↓Mcl-1L ↓Bcl-2	[32,33]
Morin		*Maclura pomifera, Psidium guajava*		Human colorectal cancer cell line (HCT-116)	48 h	350 μg/mL	↑caspase-8, caspase-9 and caspase-3, ↑Bax ↓bcl-2 ↑ROS production	[33]
				Myocardial ischemia-reperfusion injury-induced rat mofel	24 h	20 mg/kg	↓cytochrome c ↓caspase-9 and caspase-3↑Bax	[34]
Tamarixetin		*Acacia nilotica, Pteridium aquilinum*		Human leukemic cell lines (HL-60 and U937)	24 h	30 μM	↑p21 and cyclin B1 ↑Bax ↓Bid	[35]
Rutin		*Carpobrotus edulis, Ruta graveolens*		Rat cardiomyocyte-derived cell line (H9c2)	24 h	20 μM	↓Bax ↓caspase-3, ↑Bcl-2	[36]
Icariin		*Epimedium spp (Berberidaceae)*		Human hepatoma cell line (SMMC-7721)	24 h	10 μM	↑caspase cascade ↑Bax ↓Bcl-2 ↑cytochrome c release	[37]
Acacetin		*Turnera diffusa, Robinia pseudoacacia, Betula pendula*		Human breast cancer cell line (MCF-7), Human hepatoma cell line (SMMC-7721)	24 h	100 μM, 200 μM	↑caspase-7 ↑Bax↓Bcl-2 ↑AIF release ↑ROS production	[38]
Wogonin		*Scutellaria radix*		Human hepatoma cell line (SMMC-7721)	24 h	100 μmol/L	↑Bax ↓Bcl-2	[39]
Apigenin		*Citrus japonica, Citrus sinensis*		Human lung cancer cell line (A549)	24 h	150 μM	↑Bax ↓Bcl-2 ↑caspase-8, caspase-3	[40]
Chrysin		*Oroxylum indicum, Passiflora incarnata*		Human hepatocellular carcinoma cell line (HCC)	24 h	18 μg/mL, 25 μg/mL	↑p53 ↑Bad, Bax and Bak, ↓Bcl-2	[41]
Luteolin		*Brassica oleracea, Daucus carota*		Human colon adenocarcinoma cell line (HCT-15)	48 h	20 μM, 40 μM, 80 μM	↓Wnt/β-catenin signaling pathway, Cell cycle arrest at G2/M phase by ↑Bax ↑caspase-3 ↓Bcl-2	[42]
Baicalein	combinatoial	*Scutellaria baicalensis*		Human breast cancer cell line (MCF-7)	24 h	25 μmol/L	↑Caspase cascade ↑p53 ↑Bax ↓Bcl-2	[43]
Baicalin					24 h	50 μmol/L		
Eupatorin		*Tanacetum vulgare, Orthosiphon stamineus*		Human Epithelial cells (HeLa cell line)	12 h	20 μM	↑caspase cascade ↑p53 ↑p21 ↑Bax	[44]
Sinensetin		*Orthosiphon stamineus*		Human T-cell lymphoma (Jurkat cells)	12 h, 24 h	50 μmol/L, 100 μmol/L	Induce autophagy ↑caspases-3, caspase-8, caspase -9 ↑ROS production ↓Akt/mTOR signaling	[45]
Nobiletin		*Citrus nobilis*		Human breast cancer cell line (MCF-7)	24 h	100 μM	↑p53 ↑Bax ↓Bcl-2 ↓MMP-2 and MMP-9	[46]
Eupatilin		*Artemisia asiatica*		Rat adrenal phaeochromocytoma cell line (PC 12), Rat cardiomyocyte-derived cell line (H9c2)	24 h	10 μM	↓Apoptosis ↓Bax ↓caspase-3 ↑Bcl-2 ↑Akt/GSK-3β pathway	[47,48]
Vitexin		*Crataegus pinnatifida*		Human non-small-cell lung cancer cell line (A549)	48 h	20 μM, 40 μM	↑Apoptosis ↑Bax ↓Bcl-2 ↑cytochrome c release	[49]
				Rat myocardial cells			↓Apoptosis ↓Bax ↑Bcl-2	[50]
Pectolinarigenin		*Clerodendrum phlomidis, Cirsium chanroenicum, Eupatorium odoratum*		Rat osteosarcoma cells	24 h	20 μM, 50 μM	↓cell proliferation ↑apoptosis ↓cyclin D1 and survivin ↓Bcl-2 ↓Bcl-xL	[51]
				Spinal cord injury induced rat model	48 h	50 mg/kg body wt.	↓apoptosis ↓caspase-3, caspase-9 ↓Bax ↑Bcl-2 ↓PARP	[52]
Morusin		*Morus nigra*		Human prostate cancer, hepatocellular carcinoma (HCC)	24 h	30 μM	↓ STAT3 signaling ↓Bcl-2, Bcl-xL and survivin ↑caspase-3	[53]
Vicenin-2		*Citrus sinensis*		Nitrosamine-induced liver carcinoma rat model	48 h	30 mg/kg	↓ROS production ↓Bcl-xL and Bcl-2 ↑Bax	[54]
				Human colorectal adenocarcinoma cell line (HT-29)	24 h	40 μM	↑Cell cycle arrest at G2/M phase	[55]
Hydroxygenkwanin		*Daphne genkwa*		Glioma cells	24 h	25 μM	↓cell proliferation ↑TNF-α ↑caspase-3, caspase-8 ↓Bcl-2	[55]
Puerarin		*Radix puerariae*		Rat adrenal phaeochromocytoma cell line (PC 12)	12 h	50 μM	↓Bax ↑Bcl-2 ↓apoptosis	[56]
Genistein		*Flemingia vestita*		Human breast cancer cell line (MCF-7)	16 h	50 μM	↑Bax↓Bcl-2 ↓Akt phosphorylation	[57]
Daidzein		*Glycine max*		Human hepatic adenocarcinoma cell line (SK-HEP-1)			↑apoptosis ↑Bak ↓Bcl-2 ↓Bcl-xL	[58]
Tectorigenin		*Belamcanda chinensis*		Human keratinocytes (HaCaT)	24 h	1 μM, 10 μM	↓ROS production ↑Bcl-2 ↑glutathione and catalase	[59]
Jaceosidin		*Artemisia princeps*		Human primary glioblastoma cell line (U87)	24 h	100 μM	↑cell cycle arrest at G2/M phase ↑p53 ↑Bax ↑caspase-3 ↑Cytochrome c	[60]
Hesperetin		*Citrus sinensis, Citrus aurantium*		Human breast cancer cell line (MCF-7)	24 h	200 μM, 400 μM	↑cell cycle arrest at GI-phase ↑p21 ↓cyclin D1 ↓Bcl-2 ↑Bax	[60]
Hesperidin		*Citrus aurantium*		Human colon cancer cell line (SNU-C4)	24 h	10 μM, 100 μM	↑caspase-3 ↓Bcl-2 ↑Bax	[61]
Liquiritigenin		*Glycyrrhiza inflate, Glycyrrhiza uralensis, Glycyrrhiza glabra*		Rat insulinoma cell line (INS-1)	24 h	5 μM	↓apoptosis ↓caspases ↓PARP ↑Bcl-2	[62]
Naringin		*Citrus paradise, Citrus bergamia*		Gentamicin-induced nephrotoxicity rat model	24 h	100 mg/kg	↓caspase-3 ↓p53 ↓Bax ↑Bcl-2	[63]
Poncirin		*Citrus aurantium, Citrus paradise, Citrus bergamia*			24 h	130 μM	↑FasL ↑caspase-8 ↑caspase-3 ↑PARP	[64]
Kuarinone		*Sophora flavescens Aiton*		Human non-small cell lung cancer cell line (A549)	24 h	20 mg/kg, 40 mg/kg	↓Cell proliferation ↓Bcl-2 ↑Bax	[65]
Isoliquiritigenin		*Allium ascalonicum L, Glycine max L*		Human colorectal cancer (HT-29) and human cervical carcinoma cell line (Ca Ski)	24 h	40 μM	↓NO production ↓PGE2 ↓Cox-2 ↓iNOS ↑p53 ↑p21 ↑Bax ↓Bcl-2 ↑caspase cascade ↑Fas ligand expression ↑caspase-8	[66]
Isoliquiritin		*Glycyrrhiza glabra, Glycyrrhiza uralensis*		Human colorectal cancer (HT29) cells	24 h	100 μg/mL	↑p53 ↑p21 ↓Bcl-2 ↓MDM2 ↓p-Akt proteins	[67]
Licochalcone A		*Glycyrrhiza glabra, Glycyrrhiza uralensis, Glycyrrhiza inflate*		Human breast cancer cell line (MCF-7) and human leukemia cell line (HL-60)	72 h	25 μM	↑apoptosis ↑Bax ↓Bcl-2	[68]
Licochalcone B		*Glycyrrhizae radix Glycyrrhiza glabra*		Human malignant bladder cancer cell lines (T24 and EJ), murine bladder cancer cell line (MB49 tumor model)	72 h	40 μM, 80 μM /160 μM	↑apoptosis ↑Bax ↓Bcl-2 ↑caspase-3 ↑PARP	[69]
Licochalcone E		*Glycyrrhiza glabra, Glycyrrhiza inflate, Glycyrrhiza uralensis*		Human pharyngeal squamous carcinoma cell line (FaDu)	24 h	50 μM	↑apoptosis ↑FasL ↑caspase-8 and caspase-9 ↑p53 ↑Bax ↓Bcl-2	[70]
Malvidin		*Myrica rubra, Vitis vinifera, Vaccinium corymbosum*		Human colorectal cancer cells (HCT-116), rat squamous cell carcinoma cell line (SCC131)	24 h	62 μM, 70 μM	↑apoptosis ↓cell proliferation ↑p21 ↑Bax ↓Bcl-2 ↑caspase cascade	[71,72]
Cyanidin-3-O-β-glucopyranoside		*Glycine max, Hibiscus sabdariffa*		Human leukemic cell line (HL-60)	24 h	200 μg/mL	↑apoptosis ↑p53 ↑Bax ↓Bcl-2	[73]
Pelargonidin		*Rubus idaeus, Vaccinium subg. Oxycoccus*		Human colon cancer cell line (HT29)	24 h	(GI50) 0.31 μM	↑Bax ↑Bid ↓Bcl-2 ↓Bcl-xL ↑p53 ↑p21	[74]
Delphinidin		*Raphanus sativus, Phaseolus vulgaris, Solanum melongena*		Human breast cancer cell line (MDA-MB-231)	48 h	20 μmol/L	↓Akt signaling pathway ↓cell proliferation ↑apoptosis ↑Bax ↓Bcl-2	[75]
Delphinidin-3-O-glucoside		Bilberry extract	*Vaccinium myrtillus L, Triticum aestivum L, Vaccinium myrtillus*	Peripheral blood mononuclear cells (PMBCs)	24 h	30 μM, 100 μM	↑apoptosis ↓Akt signaling ↑caspase-3 ↓Bcl-2	[76]
Delphinidin-3-O-rutinoside			*(Coffea arabica L*		24 h	30 μM, 100 μM		
Epigallocatechin-3-gallate		*Camellia sinensis, Prunus avium, Prunus persica*		Humangastric cancer cell line (MKN45) and human colon cancer cell line (HT-29)	48 h	80 μM, 100 μM	↑apoptosis ↓Bcl-2 ↓COX-2 ↑AMPK ↓VEGF	[77,78,79]
Catechin		*Camellia assumica, Camellia sinensis*		Human breast cancer cell line (MCF-7) and human non-small-cell lung cancer cell line (A549)	72 h	500 μM	↑Apoptosis ↑Bax ↓Bcl-2	[80]
Procyanidin		*Pinus koraiensis*		HeLa cell line	72 h	250 µg/mL	↓Bcl-2 ↓survivin ↑Bax expression	[81]
Epicatechin		*Euonymus alatus*		C57BL/6J mice	48 h	0.50 mg/kg	↑Bcl-2 ↓Caspase-3 ↓Bax expression	[82]
Miquelianin		*Nelumbo nucifera*		SH-SY5Y cell line	24 h	200 µM	↑Bcl-2 ↓PARP ↓Bax expression	[83]
Galangin		*Alpinia officinarum*		MGC 803 cell line	24 h and 48 h	20 µM	↓Bcl-2 ↓caspase-3 ↑PARP	[84]

↑ Upregulation and ↓ Downregulation.

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
