# Peer review of "Bcl-2 Modulation in p53 Signaling Pathway by Flavonoids: A Potential Strategy towards the Treatment of Cancer"

_ijms, 2021, doi:10.3390/ijms222111315_

Round 1

Reviewer 1 Report

Review of Manuscript Number:  ijms-1374960; Title:  Bcl-2 modulation in p53 signaling pathway by flavonoids: A potential strategy towards the treatment of cancer.

Authors: Noor Rahman , Haroon Khan * , Asad Zia , Asifullah khan , Sajad Fakhri , Michael Aschner , Karim Gul , Luciano Saso

General comment: the review reports the effect of flavonoids on Bcl-2 modulation mediated by p53 signaling pathway and its interest as a potential strategy in cancer treatment. It is a very interesting update on the subject of modulation of apoptosis in cancer treatments by bioactive compounds. The review is rather complete and well organized and references quoted are reasonably updated. Some specific comments to improve the information are detailed below:

Specific comments:

  • First paragraph of introduction; text is too generic, it seems unnecessary to remind the importance of cancer in current society.
  • Lines 43-44; two promising too close in the same sentence.
  • Table 1; first flavonoid should be quercetin.
  • Table 1, quercetin in vitro; the authors should include article by Granado-Serrano et al. (2006) J. Nutrition 136:2715-2721. The results should also be discussed in epigraph 6.1.
  • Table 1, kaempferol and tamarixetin; plant genus and species should be in italics.
  • Table 1, fisetin, morin, chrysin, sinensetin, isoliquiritin, Cyanidin-3-O-β-glucopyranoside ; mL should be in capital L.
  • Table 1; baicalin, remove bold of 24 h.
  • Table 1, licochalcone B; instead of the pharmaceutical name Radix Glycyrrhizae, perhaps the authors should consider to use the three more common Glycyrrhiza species, as they do in licochalcone E.
  • Table1; to complete the review the authors should include effects and mechanisms of epicatechin from article by Granado-Serrano et al. (2007) J. Agric. Food Chem. 55:2020-2027. The results should be discussed in epigraph 6.7.
  • Lines 77-79; one of the main subfamily of flavonoids, flavanols or flavan-3-ols, is missing and should be included.
  • Lines 90-91; text unclear, it should be revised.
  • Figure 1, epicatechin, perhaps the most abundant and bioactive flavanol in nature, should be included. Also procyanidins should be mentioned in the figure. The authors should consider that while 14 flavones are depicted in the figure, only two flavanols, which are more abundant and more bioactive, are included.
  • Figure 2 and table 1; some chemical structures depicted in figure 2 are not mentioned in table 1, such as michelianin (not even in figure 1) and galangin. The structures should represent flavonoids reported in the table.
  • Figure 2; it should say casticin.
  • Line 178; this generic statement about icariin inhibiting growth of many tumor cells should be supported by a reference.
  • Lines 226 and 232; since the epigraph deals with flavones, it is not necessary to repeat the class these compounds belong to. The same applies to lines 277-290 with isoflavones.
  • Figure 4; it should say genistein.
  • Reference 28 from 2019 should be already published, no longer in press.

Author Response

Dear Editor,

On behalf of all authors, I would like to thank you for considering our manuscript for review process. We have considered all the comments given by the worthy reviewers also many thanks to all the reviewers for their corrections and helpful suggestions. Changes made in the revised manuscript in response to the comments are shown in track changes. The responses to the comments are given in detail as following.

Reviewer 1

General comment: the review reports the effect of flavonoids on Bcl-2 modulation mediated by p53 signaling pathway and its interest as a potential strategy in cancer treatment. It is a very interesting update on the subject of modulation of apoptosis in cancer treatments by bioactive compounds. The review is rather complete and well organized and references quoted are reasonably updated. Some specific comments to improve the information are detailed below:

Specific comments:

  • First paragraph of introduction; text is too generic, it seems unnecessary to remind the importance of cancer in current society.

Authors response: Alteration was made as per suggestions.

  • Lines 43-44; two promising too close in the same sentence.

Authors response: Removed the second promising as per suggestions.

  • Table 1; first flavonoid should be quercetin.

Authors response: Corrected accordingly.

  • Table 1, quercetin in vitro; the authors should include article by Granado-Serrano et al. (2006) J. Nutrition 136:2715-2721. The results should also be discussed in epigraph 6.1.

Authors response: We have discussed the results from Granado-Serrano et al., 2006, under the subsection 6.1 as below…

Granado-Serrano et al., 2006 reported quercetin-induced apoptosis in HepG2 cell line and evaluated the modulation and expression of Bcl-x and Bax. Bcl-xL has been identified as a caspase substrate and the product of Bcl-xL cleavage, Bcl-xS, has a pro-apoptotic function. The level of Bcl-xS was elevated at all quercetin concentrations compared with controls after 18 h of incubation. Quercetin reduced the Bcl-xL:Bcl-xS ratio, which reached a minimum at 50 µmol/L. Bax has been demonstrated to translocate from the cytoplasm to the outer mitochondrial membrane, where it forms holes and mediates apoptosis. Western blot analysis of mitochondrial and cytoplasmatic fractions revealed that after 18 hours of quercetin treatment, translocation of Bax to the mitochondria increased to its greatest level at 50 mmol/L, decreased at 75 mmol/L, and returned to control levels at 100 mmol/L.

  • Table 1, kaempferol and tamarixetin; plant genus and species should be in italics.

Authors response: Italicised accordingly.

  • Table 1, fisetin, morin, chrysin, sinensetin, isoliquiritin, Cyanidin-3-O-β-glucopyranoside ; mL should be in capital L.

Authors response: Corrected as per suggestions.

  • Table 1; baicalin, remove bold of 24 h.

Authors response: Removed bold style accordingly.

  • Table 1, licochalcone B; instead of the pharmaceutical name Radix Glycyrrhizae, perhaps the authors should consider to use the three more common Glycyrrhiza species, as they do in licochalcone E.

Authors response: Dear Sir, thank you so much, we have corrected the plant name (Glycyrrhizae radix).

  • Table1; to complete the review the authors should include effects and mechanisms of epicatechin from article by Granado-Serrano et al. (2007) J. Agric. Food Chem. 55:2020-2027. The results should be discussed in epigraph 6.7.

Authors response: We have added the effects and mechanisms of epicatechin under subsection 6.7. The text is produced as below…

Epicatechin is one of the most abundant flavonoids found in apples, grapes, blackberries, etc.  A research study suggested that Epicatechin (EC) extracted from Euonymus alatus had a protective effect against acute liver injury of mice by inhibiting apoptosis in hepatocytes. Western blot analysis revealed reduced expression of cleaved Caspase-3 and Bax. Their results revealed it increased the expression of Bcl-2, confirming its protective effect in liver injury.

  • Lines 77-79; one of the main subfamily of flavonoids, flavanols or flavan-3-ols, is missing and should be included.

Authors response: We have included the subclass flavanols.

  • Lines 90-91; text unclear, it should be revised.

Authors response: The sentence is revised and clearer now, the text is reproduced below

On the other hand, Bcl-2 Associated X (Bax) protein with similar sequence and structure homology to Bcl-2, that heterodimerize with Bcl-2 induce apoptosis.

  • Figure 1, epicatechin, perhaps the most abundant and bioactive flavanol in nature, should be included. Also procyanidins should be mentioned in the figure. The authors should consider that while 14 flavones are depicted in the figure, only two flavanols, which are more abundant and more bioactive, are included.

Authors response: We have included epicatechin and procyanidins as per suggestions.The tex is produced as below….

Epicatechin is one of the most abundant flavonoids found in apples, grapes, blackberries, etc.  A research study suggested that Epicatechin (EC) extracted from Euonymus alatus had a protective effect against acute liver injury of mice by inhibiting apoptosis in hepatocytes. Western blot analysis revealed reduced expression of cleaved Caspase-3 and Bax. Their results revealed it increased the expression of Bcl-2, confirming its protective effect in liver injury.

Procyanidins are formed from the oligomers of catechin and epicatechin. When they are depolymerized in an oxidative environment, cyanidins are formed. Procyanidins extracted from Pinus koraiensis bark promoted apoptosis in the HeLa cell line by raising Bax protein expression and inhibiting Bcl-2 and survivin protein expression.

  • Figure 2 and table 1; some chemical structures depicted in figure 2 are not mentioned in table 1, such as michelianin (not even in figure 1) and galangin. The structures should represent flavonoids reported in the table.

Authors response: Dear Reviewer thank you so much for pointing out the missing flavonols and we have included these flavonoids accordingly.

  • Figure 2; it should say casticin.

Authors response: Corrected accordingly.

  • Line 178; this generic statement about icariin inhibiting growth of many tumor cells should be supported by a reference.

Authors response: Reference inserted as per suggestions

  • Lines 226 and 232; since the epigraph deals with flavones, it is not necessary to repeat the class these compounds belong to. The same applies to lines 277-290 with isoflavones.

Authors response: Removed the repetitions accordingly.

  • Figure 4; it should say genistein.

Authors response: Corrected accordingly

  • Reference 28 from 2019 should be already published, no longer in press.

Authors response: Reference updated as per suggestions.

Reviewer 2 Report

The paper “Bcl-2 modulation in p53 signaling pathway by flavonoids: A potential strategy towards the treatment of cancer” by Rahman et al. discusses the effect of a wide plethora of flavonoids against Bcl2 via the p53 pathway. The article is thorough and Authors dealt with everything with a proper level of detail. Before consideration for publication, the Authors need to address some points:

1) The English language needs to be properly and deeply checked throughout the manuscript, due to many errors present (i.e., repeated words, missing pronouns, wrong verb tenses, etc.).

2) Line 50, what is BH3? Please, specify the acronym.

3) In Table 1, what does G150 mean? Moreover, the term dose should be juxtaposed with concentration, due to the presence of in vitro studies too.

4) Throughout the text and in table, please check if all the genus and species names of plants are in italics.

5) In my opinion, Figure 1 would have been more appropriate in section 3 rather than 6.

6) Figures from 2 to 5 should be checked in order to have enough spaces among structures, to have the same color (i.e., no red groups), not to have the letter defining the rings and so on. Moreover, in the relative captions, the word “that” should be removed.

7) Why did not Authors discuss also the effect of extracts rich of flavonoids on Bcl-2 and p53? There are plenty of articles dealing with flavonoid-rich extract and this pathway. Therefore, Authors should at least discuss the most relevant ones.

Author Response

Dear Editor,

On behalf of all authors, I would like to thank you for considering our manuscript for review process. We have considered all the comments given by the worthy reviewers also many thanks to all the reviewers for their corrections and helpful suggestions. Changes made in the revised manuscript in response to the comments are shown in track changes. The responses to the comments are given in detail as following.

Reviewer 2

Comments and Suggestions for Authors

The paper “Bcl-2 modulation in p53 signaling pathway by flavonoids: A potential strategy towards the treatment of cancer” by Rahman et al. discusses the effect of a wide plethora of flavonoids against Bcl2 via the p53 pathway. The article is thorough and Authors dealt with everything with a proper level of detail. Before consideration for publication, the Authors need to address some points:

  • The English language needs to be properly and deeply checked throughout the manuscript, due to many errors present (i.e., repeated words, missing pronouns, wrong verb tenses, etc.).

Authors response: We have corrected the grammatical and typos errors as shown in track changes.

  • Line 50, what is BH3? Please, specify the acronym.

Authors response: We have added the full form of BH3 (Bcl-2 Homology 3).

  • In Table 1, what does G150 mean? Moreover, the term dose should be juxtaposed with concentration, due to the presence of in vitrostudies too.

Authors response: G150 was written mistakenly and we have corrected it to GI50 (the concentration that causes 50% growth inhibition). And dose were juxtaposed with concentration.

  • Throughout the text and in table, please check if all the genus and species names of plants are in italics.

Authors response: We have italicised all the genus and species names of plants.

  • In my opinion, Figure 1 would have been more appropriate in section 3 rather than 6.

Authors response: As per the reviewer suggestions, we have placed figure 1 in section 3.

  • Figures from 2 to 5 should be checked in order to have enough spaces among structures, to have the same color (i.e., no red groups), not to have the letter defining the rings and so on. Moreover, in the relative captions, the word “that” should be removed.

Authors response: We have checked the figures and changes the red colour, removed letters from the figures except Quercetin because these rings are discussed in section 3, and also removed “that” from all captions.

  • Why did not Authors discuss also the effect of extracts rich of flavonoids on Bcl-2 and p53? There are plenty of articles dealing with flavonoid-rich extract and this pathway. Therefore, Authors should at least discuss the most relevant ones.

Authors response: Dear Sir, thank you so much for your kind suggestions but the current review focuses specifically on flavonoids. If we add about flavonoids enriched plant extracts, it will be beyond the scope of this review.

Reviewer 3 Report

The article by Rahman and co-workers, entitled “Bcl-2 modulation in p53 signaling pathway by flavonoids: A potential strategy towards the treatment of cancer” gathers the scientific evidence on the effect of several classes of flavonoids in interfering the Bcl2/p53 pathway as a possible strategy against cancer. Overall, the article gives a complete view on the chosen subject. However, few major points need to be addressed before the paper can be considered for publication:

  • In the Abstract, Authors specifically talk about the relevance of finding novel multi-target therapies in cancer. I definitely agree with the Authors, however, in the body of the article, I have not found any mention to this. Although each flavonoid can act as multi-target agent, this is true mainly when it is present in phytocomplexes (i.e., extracts), because different compounds within the same mixture hit contemporarily different targets, achieving a multi-target therapy. Therefore, I suggest Authors to discuss the relevance of flavonoid-rich extracts (i.e., those from Citrus or berry juices) in the context of Bcl-2/p53-mediated cancer.
  • What does Author mean with the sentence in lines 9-11? Please make it clearer.
  • Which are the Bcl-2 inhibitors currently employed in clinical settings? It would be a nice improvement to expand lines 55-56.
  • Table 1 collects all the references discussed in the rest of the manuscript. Therefore, this should be specified in the text to help the reader better understand its meaning. Moreover, Table 1 would have been better in my opinion if set horizontally, without the column “Types of study” since defining the next one as “Cell Line(s)/Animal Model(s)”, the type of study discussed would be clear.
  • In line 121, what did Authors mean by saying “…activates the production of apoptotic mediators”. Please be more precise not to say something wrong.
  • Which are the “acute toxic effects” pointed out in line 375?
  • In the Conclusions, the paragraph from line 398 to 404 is redundant since repeats what is written just the section above.
  • In Figure 6, in the caption “showing” should be “show” and “led” should be “lead”.

Author Response

Dear Editor,

On behalf of all authors, I would like to thank you for considering our manuscript for review process. We have considered all the comments given by the worthy reviewers also many thanks to all the reviewers for their corrections and helpful suggestions. Changes made in the revised manuscript in response to the comments are shown in track changes. The responses to the comments are given in detail as following.

Reviewer 3

Comments and Suggestions for Authors

The article by Rahman and co-workers, entitled “Bcl-2 modulation in p53 signaling pathway by flavonoids: A potential strategy towards the treatment of cancer” gathers the scientific evidence on the effect of several classes of flavonoids in interfering the Bcl2/p53 pathway as a possible strategy against cancer. Overall, the article gives a complete view on the chosen subject. However, few major points need to be addressed before the paper can be considered for publication:

Authors response: Dear Sir,  thank you so much for the compliment, and your valuable comments and suggestions improved the overall quality of the manuscript.

  • In the Abstract, Authors specifically talk about the relevance of finding novel multi-target therapies in cancer. I definitely agree with the Authors, however, in the body of the article, I have not found any mention to this. Although each flavonoid can act as multi-target agent, this is true mainly when it is present in phytocomplexes (i.e., extracts), because different compounds within the same mixture hit contemporarily different targets, achieving a multi-target therapy. Therefore, I suggest Authors to discuss the relevance of flavonoid-rich extracts (i.e., those from Citrus or berry juices) in the context of Bcl-2/p53-mediated cancer.

Authors response: Dear Sir, thank you so much for your suggestions but the current review focuses only on flavonoids while the flavonoid-rich extracts will also contain alkaloids, terpenoids, tannins etc and if we add about extracts then it will be beyond the scope of this review.

  • What does Author mean with the sentence in lines 9-11? Please make it clearer.

Authors response: We have rewritten the sentence and clearer now and text is reproduced as below…

Flavonoids inhibit B-cell lymphoma 2 (Bcl-2) via the p53 signalling pathway, which is a significant apoptotic target in many cancer types, hence suppressing a major dysregulated pathway in cancer.

  • Which are the Bcl-2 inhibitors currently employed in clinical settings? It would be a nice improvement to expand lines 55-56.

Authors response: We have added about the Bcl2 inhibitors and the text produce as below..

In recent years, several BCL-2 protein inhibitors have been developed (Oblimersen, ABT-737, ABT-263, obatoclax mesylate, AT-101, and S55746), to specifically target BCL-2 protein for cancer treatments.

  • Table 1 collects all the references discussed in the rest of the manuscript. Therefore, this should be specified in the text to help the reader better understand its meaning. Moreover, Table 1 would have been better in my opinion if set horizontally, without the column “Types of study” since defining the next one as “Cell Line(s)/Animal Model(s)”, the type of study discussed would be clear

Authors response: We have double checked the revised manuscript and these information are already discussed in detail in the text. The table 1 of the revised manuscript has been formatted according to the reviewer suggestions.

  • In line 121, what did Authors mean by saying “…activates the production of apoptotic mediators”. Please be more precise not to say something wrong.

Authors response: We have revised the sentence as below..

In turn, Bax binds to Bcl-2, thereby activating the production of apoptotic mediators (e.g., caspase 3/9 and cytochrome C).

  • Which are the “acute toxic effects” pointed out in line 375?

Authors response: We have added the acute toxic effects of flavonoids consumption and the texts is reproduced as below…

Although most flavonoids/phenolics are considered safe, flavonoid/phenolic therapy or chemopreventive use should be evaluated because there have been reports of toxic flavonoid-drug interactions, liver failure, hemolytic anaemia, contact dermatitis, and estrogenic-related concerns such as male reproductive health and breast cancer linked to dietary flavonoid/phenolic consumption or exposure.

  • In the Conclusions, the paragraph from line 398 to 404 is redundant since repeats what is written just the section above.

Authors response: We have removed the redundancy and retained and rewritten the following sentence….

Flavonoids are degraded by intestinal microorganisms/enzymes after being administered orally. Thus, developing innovative flavonoid delivery strategies may enhance their anticancer properties.

  • In Figure 6, in the caption “showing” should be “show” and “led” should be “lead”.

Authors response: Corrected accordingly.

Reviewer 4 Report

The topic of this review is ver interesting, but the manuscript has been carelessly prepared and lacks the standard of a good authoritative review. I cannot see a proper justification why this study is important if authors does not advocate strongly for same.

The topic has not been analyzed properly as a detailed methodology section is missing. How was the data mining done? How many papers were screened? How many included? What were the keywords and which databases were used to select papers? 

Moreover, moderate english changes are required.

 I also have some suggestions, as follow below:

Line 29: “Cancer is is”

Lines 34-36:  “The World Health Organization (WHO) has estimated that 84 million people died from cancer 35 between 2005 and 2015 [6].” This reference is not the best option for this sentence. I suggest using the WHO reference and bring up the stimative for the next years.

Table 1: I suggest: “Effective Dose (in vivo) or GI50, IC50, EC50 (in vitro)” – please note that GI50 is written with letter I instead of number 1.

Table 1: Please ajust the table to better fit the titles into the content. Please standardize H460 as NCI-H460 and also the cancer nomenclature, example: human non-small cell lung cancer cell line or human non-small-cell lung carcinoma cell line / human colon cancer cells or human colorectal cancer cells... etc..

Lines 67-69: The sentence Furthermore, with the increased recognition on natural products in cancer therapy (Table 1), only 1/5–1/6 of  plant species have been screened for their medical applications [27].” seems to be na adversity phrase. Please rewrite it.

Lines 81-83: Please consider “In this line, Down-regulation of mutant p53, arresting cell cycle, inhibition of tyrosine kinase/heat shock proteins, Ras protein inhibition and binding to estrogen receptor are the most common mechanisms of flavonoids in combating cancer [31]”

Lines 89-92: Please, rewrite this sentence: “While Bcl-2 prevents programmed cell death pro-apoptotic protein Bcl-2 Associated X (Bax) on ther hand, even with similar sequence and structure homology to Bcl-2, that heterodimerize with Bcl-2 increases cell death”

Lines 100-104: Is this paragraph really necessary? It has been earlier reported that the Bcl-2 expression of malignant/normal of the myeloid lineage cells produced significant effects [36]. The authors used an anti-Bcl-2 monoclonal antibody (Mab) for the analysis, which was specific for the 26-kD protein and confirmed by Western blot analysis of a variety of myeloblastic leukaemia’s. It was also confirmed the normal size Bcl-2 of 7.5 kb, in vitro (HL-60 and KG1 cell lines) through Northern blot analysis [37].

Line 120: “thereby  activatING”

Line 122: “offer an eficient means”

Figure 2: “Selected chemical structures of flavonols that targeting Bcl-2 in the p53 pathway.”

Lines 144-145: please rewrite: “Kaempferol treatment reserved the progression of the tumor xenografts, it downregulated cyclin B1 and Cdk1 proteins.”

Line 147: chemoprEventive

Line 157: “Consistently, morin, is a bioflavonoid found in the mulberry, acts as an apoptotic inducing agent in multiple human cancers.

Lines 173-175: The sentence does not make sense: “H2O2, a ROS that initiates the caspase-3 activation and enhanced Bax expression while reducing Bcl-2 expression and leading to cell death in rats H9c2 cell lines”

Mentions to figures 2, 3, 4 and 5 should cover all the compounds mentioned, not exclusively after citing 1 specific OR one sentence refering those figures shouldbe included in the beginning at each section.

Line 250: Another flavonoid glycoside, vicenin-2 treatment reversed the diethyl nitrosamine.....

Line 285: “isoangustone A, (please remove comma) induced apoptosis”

Line 321: CaSki

Line 366: please rewrite: “The anti-apoptotic effects of catechin and gemigliptin on tacrolimus-induced renal injury in mice [113].”

Figure 6 title: “It showing details of various events including 396 down/up regulation or expression that led to apoptosis” please rewrite it.

Figure 6: DNA fragmentation instead of fragmenttion.

Author Response

Dear Editor,

On behalf of all authors, I would like to thank you for considering our manuscript for review process. We have considered all the comments given by the worthy reviewers also many thanks to all the reviewers for their corrections and helpful suggestions. Changes made in the revised manuscript in response to the comments are shown in track changes. The responses to the comments are given in detail as following.

Reviewer 4

Comments and Suggestions for Authors

The topic of this review is very interesting, but the manuscript has been carelessly prepared and lacks the standard of a good authoritative review. I cannot see a proper justification why this study is important if authors do not advocate strongly for same.

 Author response: Yes, we do agree with the reviewers’ comments regarding the lack of clear objectives of this manuscript we are indebted to the respected reviewer for pointing out this important point. The objectives/justification of this study has now been inserted at the end of abstract section of the revised manuscript as…

To date, there have been no studies reported which extensively highlight the role of flavonoids and especially the different classes of flavonoids in the modulation of Bcl-2 in P53 signaling pathway. Herein, we discuss the modulation of Bcl-2 in p53 signaling pathway by different classes of flavonoids and highlight different mechanisms through which this modulation can occurs. This study will provide a rationale for the use of flavonoids against different cancers paving a new mechanistic-based approach to cancer therapy.

The topic has not been analyzed properly as a detailed methodology section is missing. How was the data mining done? How many papers were screened? How many included? What were the keywords and which databases were used to select papers? 

Authors response: We agree with the reviewers’ comment regarding the lacking of detailed methodology on data collection However, this is a traditional literature review rather than a meta-analysis. We used different search engines like pubmed, google scholar, Scopus and Sci-finder and collected related papers for each section from 1992-2021.

Moreover, moderate english changes are required.

 Authors response: We have improved the language and showed in track changes.

 I also have some suggestions, as follow below:

 Line 29: “Cancer is is”

Authors response: Removed accordingly.

Lines 34-36:  “The World Health Organization (WHO) has estimated that 84 million people died from cancer 35 between 2005 and 2015 [6].” This reference is not the best option for this sentence. I suggest using the WHO reference and bring up the stimative for the next years.

 Authors response: We have updated the cancer statistics by following “Global Cancer Statistics 2020: GLOBOCAN Estimates of Incidence and Mortality Worldwide for 36 Cancers in 185 Countries”.

And the text were produced as below..

Worldwide, an estimated 19.3 million new cancer cases (18.1 million excluding nonmelanoma skin cancer) and almost 10.0 million cancer deaths (9.9 million excluding nonmelanoma skin cancer) occurred in 2020. The global cancer burden is expected to be 28.4 million cases in 2040, a 47% rise from 2020, with a larger increase in transitioning (64% to 95%) versus transitioned (32% to 56%) countries due to demographic changes, although this may be further exacerbated by increasing risk factors associated with globalization and a growing economy.

Table 1: I suggest: “Effective Dose (in vivo) or GI50, IC50, EC50 (in vitro)” – please note that GI50 is written with letter I instead of number 1.

 Authors response: Changed as per suggestions and also corrected GI50.

Table 1: Please ajust the table to better fit the titles into the content. Please standardize H460 as NCI-H460 and also the cancer nomenclature, example: human non-small cell lung cancer cell line or human non-small-cell lung carcinoma cell line / human colon cancer cells or human colorectal cancer cells... etc..

Authors response: We have adjusted the table accordingly and standardized the nomenclature of cell lines.

Lines 67-69: The sentence “Furthermore, with the increased recognition on natural products in cancer therapy (Table 1), only 1/5–1/6 of  plant species have been screened for their medical applications [27].” seems to be na adversity phrase. Please rewrite it.

 Authors response: The sentence was revised and reproduced as below…

In addition, despite the increased recognition of natural products in cancer therapy presented in Table 1, Only about one-fifth to one-sixth of plant species have been evaluated for medical purposes  

Lines 81-83: Please consider “In this line, Down-regulation of mutant p53, arresting cell cycle, inhibition of tyrosine kinase/heat shock proteins, Ras protein inhibition and binding to estrogen receptor are the most common mechanisms of flavonoids in combating cancer [31]”

 Authors response: Alteration was made accordingly.

Lines 89-92: Please, rewrite this sentence: “While Bcl-2 prevents programmed cell death pro-apoptotic protein Bcl-2 Associated X (Bax) on ther hand, even with similar sequence and structure homology to Bcl-2, that heterodimerize with Bcl-2 increases cell death”

Authors response: We have revised the sentence as below…

On the other hand, Bcl-2 Associated X (Bax) protein with similar sequence and structure homology to Bcl-2, that heterodimerize with Bcl-2 induce apoptosis.

Lines 100-104: Is this paragraph really necessary? It has been earlier reported that the Bcl-2 expression of malignant/normal of the myeloid lineage cells produced significant effects [36]. The authors used an anti-Bcl-2 monoclonal antibody (Mab) for the analysis, which was specific for the 26-kD protein and confirmed by Western blot analysis of a variety of myeloblastic leukaemia’s. It was also confirmed the normal size Bcl-2 of 7.5 kb, in vitro (HL-60 and KG1 cell lines) through Northern blot analysis [37].

Authors response: We have removed the unnecessary sentence and make the paragraph as below…

It has been earlier reported that the Bcl-2 expression of malignant/normal myeloid lineage cells produced significant effects [36]. Overexpression of the Bcl-2 gene is inherent to various cancers, for example, 90% of colorectal adenocarcinomas, 80% of undifferentiated nasopharyngeal cancers, 70% of breast adenocarcinomas and chronic lymphocytic leukaemias, 60% of gastric cancers, 30-60% of prostate cancers and in varying percentages of melanomas, blastomas and kidney cancers [38].

Line 120: “thereby  activatING”

Authors response: Corrected as per suggestions.

Line 122: “offer an eficient means”

Authors response: Removed “an” accordingly.

Figure 2: “Selected chemical structures of flavonols that targeting Bcl-2 in the p53 pathway.”

Authors response: Dear Sir, the second reviewer suggested to remove “that” from the captions and I think the caption is now clearer ( Selected chemical structures of flavonols targeting Bcl-2 in the p53 pathway).

Lines 144-145: please rewrite: “Kaempferol treatment reserved the progression of the tumor xenografts, it downregulated cyclin B1 and Cdk1 proteins.”

Authors response: The sentence has been rewritten and the text below.

Kaempferol therapy slowed the progression of tumor xenografts by downregulating the proteins cyclin B1 and Cdk1.

Line 147: chemoprEventive

Authors response: Corrected accordingly.

Line 157: “Consistently, morin, is a bioflavonoid found in the mulberry, acts as an apoptotic inducing agent in multiple human cancers.

Authors response: Alteration was made accordingly.

Lines 173-175: The sentence does not make sense: “H2O2, a ROS that initiates the caspase-3 activation and enhanced Bax expression while reducing Bcl-2 expression and leading to cell death in rats H9c2 cell lines”

Authors response: The sentence was rewritten and reproduce as below.

H2O2 is a Reactive Oxygen Species (ROS) that activates caspase-3 and increases Bax expression while decreasing Bcl-2 expression and causing cell death in rat H9c2 cell lines.

Mentions to figures 2, 3, 4 and 5 should cover all the compounds mentioned, not exclusively after citing 1 specific OR one sentence refering those figures shouldbe included in the beginning at each section.

Authors response: We have cited all figures in the text accordingly.

Line 250: Another flavonoid glycoside, vicenin-2 treatment reversed the diethyl nitrosamine.....

Authors response: Removed “treatment” as per suggestions.

Line 285: “isoangustone A, (please remove comma) induced apoptosis”

Authors response: Comma removed.

Line 321: CaSki

Authors response: Dear Sir, as per ATCC site (https://www.atcc.org/products/crl-1550) this cell name is correct (Ca Ski) and you can cross check it.

Line 366: please rewrite: “The anti-apoptotic effects of catechin and gemigliptin on tacrolimus-induced renal injury in mice [113].”

Authors response: Sentence rewritten as below..

Catechin and gemigliptin have anti-apoptotic effects on tacrolimus-induced renal injury in mice.

Figure 6 title: “It showing details of various events including 396 down/up regulation or expression that led to apoptosis” please rewrite it.

Authors response: Dear Sir Thank you so much, the second reviewer also suggested for it’s correction and we have rewritten the caption as below…

It shows details of various events including up and down-regulation or expression that leads to apoptosis.

Figure 6: DNA fragmentation instead of fragmenttion.

Authors response: Corrected accordingly

Round 2

Reviewer 3 Report

Reviewer I round: In the Abstract, Authors specifically talk about the relevance of finding novel multi-target therapies in cancer. I definitely agree with the Authors, however, in the body of the article, I have not found any mention to this. Although each flavonoid can act as multi-target agent, this is true mainly when it is present in phytocomplexes (i.e., extracts), because different compounds within the same mixture hit contemporarily different targets, achieving a multi-target therapy. Therefore, I suggest Authors to discuss the relevance of flavonoid-rich extracts (i.e., those from Citrus or berry juices) in the context of Bcl-2/p53-mediated cancer.

Authors response: Dear Sir, thank you so much for your suggestions but the current review focuses only on flavonoids while the flavonoid-rich extracts will also contain alkaloids, terpenoids, tannins etc and if we add about extracts then it will be beyond the scope of this review.

Reviewer II round: In pharmacology, when the term multi-target is employed, it is referred to the employment of different drugs contemporarily aiming at different targets, hence achieving a joined effect. When Authors claim in the Abstract the multi-target activity of flavonoids only reporting studies on the effects of single flavonoids in the rest of the text, they make an underestimation mistake. Therefore, I reiterate to the Authors the need of discussing the effect of mixture of flavonoids (see I round above) or at least, to correct the sentences at lines 17-22.

Author Response

Reviewer 3:

Reviewer II round: In pharmacology, when the term multi-target is employed, it is referred to the employment of different drugs contemporarily aiming at different targets, hence achieving a joined effect. When Authors claim in the Abstract the multi-target activity of flavonoids only reporting studies on the effects of single flavonoids in the rest of the text, they make an underestimation mistake. Therefore, I reiterate to the Authors the need of discussing the effect of mixture of flavonoids (see I round above) or at least, to correct the sentences at lines 17-22.

Authors response: We admire the respected reviewer for the comment. To prevent such mistakes, we have now rephrased the sentences (lines 17-22) in the abstract (Highlighted in red).

Reviewer 4 Report

The authors have adjusted all the points mentioned by this and other reviewers and I believe the paper is better prepared for publication now. 

One paragraph that is still not making sense for me is: "On the other hand, Bcl-2 Associated X (Bax) protein with similar sequence and structure homology to Bcl-2, that heterodimerize with Bcl-2 induce apoptosis.". Should it be: "On the other hand, Bcl-2 Associated X (Bax) protein, which has similar sequence and structure homology to Bcl-2 and heterodimerizes with it, induces apoptosis." ? 

Author Response

Reviewer 4:

The authors have adjusted all the points mentioned by this and other reviewers and I believe the paper is better prepared for publication now. 

Authors response: Thank you so much for considering our manuscript for publication.

One paragraph that is still not making sense for me is: "On the other hand, Bcl-2 Associated X (Bax) protein with similar sequence and structure homology to Bcl-2, that heterodimerize with Bcl-2 induce apoptosis.". Should it be: "On the other hand, Bcl-2 Associated X (Bax) protein, which has similar sequence and structure homology to Bcl-2 and heterodimerizes with it, induceapoptosis." ? 

Authors response: Dear reviewer, thank you so much for the correction and we have corrected the sentence accordingly and highlighted in the text.

Round 3

Reviewer 3 Report

None